# Cognitive Model Discovery via Disentangled RNNs

**Kevin J. Miller**
Google DeepMind and University College London
London, UK
kevinjmiller@deepmind.com

**Maria Eckstein**
Google DeepMind
London, UK
mariaeckstein@deepmind.com

**Matthew M. Botvinick**
Google DeepMind
London, UK
botvinick@deepmind.com

**Zeb Kurth-Nelson**
Google DeepMind and University College London
London, UK
zebk@deepmind.com

## Abstract

Computational cognitive models are a fundamental tool in behavioral neuroscience. They instantiate in software precise hypotheses about the cognitive mechanisms underlying a particular behavior. Constructing these models is typically a difficult iterative process that requires both inspiration from the literature and the creativity of an individual researcher. Here, we adopt an alternative approach to learn parsimonious cognitive models directly from data. We fit behavior data using a recurrent neural network that is penalized for carrying information forward in time, leading to sparse, interpretable representations and dynamics. When fitting synthetic behavioral data from known cognitive models, our method recovers the underlying form of those models. When fit to laboratory data from rats performing either a reward learning task or a decision-making task, our method recovers simple and interpretable models that make testable predictions about neural mechanisms.

## 1 Introduction

Fitting quantitative cognitive models to behavioral data is a fundamental tool in many areas of psychology and neuroscience [51, 12, 10, 39]. These models can be viewed as mechanistic hypotheses about the cognitive processes used by the brain. Viewed in this way, they act as an explicit software instantiation of a particular cognitive hypothesis, and can be used to make precise quantitative predictions about behavioral and neural data. A traditional modeling pipeline is for a human researcher to iterate over a three-step process: first to propose a candidate model structure (e.g. Q-learning), second to optimize model parameters (e.g. learning rate) with respect to a behavioral dataset, and finally to check whether the resulting model reproduces scientifically important features of the dataset. However, discovering an appropriate model structure is difficult (there are many possible structures to explore) as well as biased (the best structure may be one that the researcher has not thought of).

An alternative approach is fitting recurrent neural networks (RNNs) directly to behavior using supervised learning [16, 19, 46, 2]. RNNs are highly expressive and can approximate a wide variety of model structures in their weights. This is therefore a way of discovering a well-fitting model from data automatically. The drawback of this approach is that the resulting RNN is a "black box": a complex system that itself requires further analysis if it is to yield insight into cognitive mechanism.

Here, we propose a solution that aims to achieve the best of both worlds: automated discovery of human-interpretable cognitive models. Our approach, which we call disentangled RNN or "DisRNN", draws on recently developed methods from machine learning for "disentangling" [23, 24, 8, 21].

37th Conference on Neural Information Processing Systems (NeurIPS 2023).

These methods encourage networks to learn representations in which each dimension corresponds to a single true factor of variation in the data [23, 8]. DisRNN encourages disentangling in two ways. The first is to separate the update rule for each element of the latent state into separate sub-networks. The second is to use information "bottlenecks", which impose a penalty on maintaining information within the network, to the inputs and outputs of these sub-networks.

We fit DisRNN on sequential behavioral datasets from rats and artificial agents performing two classic cognitive tasks: a dynamic two-armed bandit task [13, 28, 26, 43] and a "pulse accumulation" decision-making task [7, 45, 14, 41]. First, we generate synthetic datasets for the two-armed bandit task using artificial agents with known learning algorithms (Q-Learning and Actor-Critic) which have different update rules and carry different information between timesteps. When fitting these synthetic datasets, DisRNN correctly recovers the timecourses of latent state information and the structure of the update rules. Second, we fit DisRNN on large laboratory datasets generated by rats performing the same two-armed bandit task [37]. We find that DisRNN provides a similar quality of fit to the best known human-derived cognitive model of this dataset [37] as well as to an unconstrained neural network [16]. Third, we generate a synthetic dataset for the decision-making task using an agent with a known decision algorithm (bounded accumulation) and show that disRNN is able to correctly recover the structure of its decision rule. Finally, we fit disRNN to a large laboratory dataset of rats performing this task [7], and find that it recovers a strategy similar to that of the best known human-derived model for this dataset. In all cases, we find that DisRNN is able to learn simple human-interpretable cognitive strategies which can be used to make predictions about behavioral and neural datasets.

## 2 Related Work

Our strategy for encouraging networks to adopt disentangled representations is directly inspired by work using variational autoencoders (VAE). Specifically, $\beta$-VAE [23, 8], by scaling the KL loss of a variational autoencoder's sampling step (which can be viewed as requiring information to pass through a Gaussian information bottleneck), often learns sparse, disentangled representations in which each latent variable corresponds to a single true factor of variation in the data. While $\beta$-VAE considered feedforward autoencoders, a wide variety of techniques have been proposed combining elements of feedforward VAEs with recurrent neural networks [20]. We adapt these ideas here in a way that emphasizes interpretability and is appropriate for cognitive model discovery.

The use of recurrent neural networks as cognitive models has a long history [5, 27], and recent work has expanded the toolkit for fitting networks to behavioral datasets and for interpreting them [16, 19, 46, 15]. This approach of fitting standard neural networks and working to interpret the resulting fits benefits from the growing toolkit for neural network interpretability [47, 35, 36]. Our approach is complementary: instead of fitting standard networks and developing tools to interpret them post hoc, we develop networks which are incentivized to learn easily-interpretable solutions. Within neuroscience, a number of methods exist which attempt to discover interpretable latent dynamics from neural recording data [40, 50, 29, 44]. To our knowledge, these ideas have not been applied in the context of cognitive model discovery from behavioral data.

The two-armed bandit task we consider here is one of a family of dynamic reward learning tasks that have been heavily studied in behavioral neuroscience. This has led to a large library of candidate cognitive models [13, 4, 17, 32, 11, 26, 34, 37, 33, 3, 42]. The development of such models typically follows a theory-first approach, beginning with an idea (e.g. from optimality [13, 42], from machine learning [33], or from neurobiology [34]). Several studies have fit behavioral dataset using highly flexible classical models [26, 37, 32, 11]. One of these has attempted to use these fits as a basis for a process of data-driven cognitive model discovery by compressing the revealed patterns into a simpler model which is cognitively interpretable [37].

Two very recent papers have proposed a very similar workflow of discovering cognitive strategies from behavioral data using constrained neural networks. The first uses "hybrid" models that combine elements of classic structured models and flexible neural networks [18]. The second constrains networks by limiting them to a very small number of hidden units [2]. Future work should compare these approaches directly on matched datasets, as well as explore other possible opportunities for adapting modern machine learning techniques as tools for cognitive model discovery.

# 3 Disentangled Recurrent Neural Networks

An RNN trained to match a behavioral dataset containing temporal dependencies [16, 19, 46] can be viewed as maintaining a set of *latent variables* which carry information from the past that is useful for predicting the future. The weights of the network can be viewed as defining a set of *update rules* defining how each latent variable evolves over time based on external observations as well as the previous values of the other latent variables. Standard RNN architectures (such as the GRU [9]) typically learn high-dimensional latent representations with highly entangled update rules. This makes it difficult to understand which cognitive mechanisms they have learned, limiting their usefulness as cognitive hypotheses.

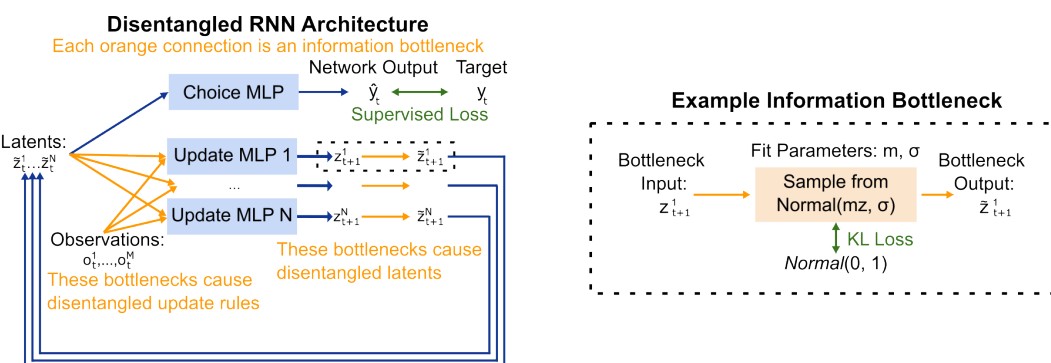

Figure 1: **Network Architecture**. **Left**: Overall architecture of the DisRNN. Each latent variable is updated by a separate feedforward neural network (MLP: Multilayer Perceptron). Bottlenecks (orange connections) are imposed both on the inputs to these networks and on each latent variable to encourage interpretable representations. **Right**: A single information bottleneck. Output $\tilde{z}$ is a random sample from a Gaussian distribution determined by input $z$ and bottleneck parameters $m$ and $\sigma$. The bottleneck is associated with a KL loss penalizing information transfer.

## 3.1 Network Architecture

In order to learn an interpretable cognitive model, we encourage sparsity and independence by imposing bottlenecks which penalize the network for using excess information (Figure 1, right) [49, 1, 31, 23, 24, 8, 21]. The bottlenecks in our networks limit information flow using Gaussian noise. Our implementation uses multiple information bottlenecks, each of which is a noisy channel defined by two learned parameters: a "multiplier" $m$ and noise variance $\sigma$. The output on each timestep is sampled from a Gaussian distribution:

$$\tilde{z}_t \sim \mathcal{N}(mz_t, \sigma) \tag{1}$$

where $x$ is the scalar input to the bottleneck and $\tilde{x}$ is the scalar output. Each bottleneck is associated with a loss which penalizes the sampling distribution for deviating from the unit Gaussian:

$$L_{\text{bottleneck}} = \sum_t D_{KL}\big(\mathcal{N}(m_b x_{b,t}, \sigma_b)||\mathcal{N}(0,1)\big) \tag{2}$$

where $D_{KL}$ is the Kullback-Leiber divergence, a measure of the difference between two probability distributions. This difference, and therefore the cost of the bottleneck, will be zero in the case that $m = 0$ and $\sigma = 1$. In that situation, the bottleneck will output samples from the unit Gaussian. These outputs will be independent of the input, meaning that no information will flow through the bottleneck. We refer to these as "closed bottlenecks". In all other situations, $L_b$ will be greater than zero, and some information will pass through. In our experiments, bottlenecks that are carrying useful information typically fit $m \approx 1$ and $\sigma \ll 1$. We refer to these as "open bottlenecks".

We use these information bottlenecks to encourage cognitively interpretable models by encouraging two distinct kinds of disentangling. The first is disentangling in the latent variables themselves: we would like these to capture separable latent processes and to be relatively few in number. We

encourage this kind of simplicity by imposing a separate bottleneck on each scalar element of the network's hidden state. We refer to these as the "latent bottlenecks".

$$\tilde{z}_t^i \sim \text{bottleneck}_i(z_t^i) \tag{3}$$

The second kind of simplicity is in the update rules for the latent variables. We would like each variable to be updated by its own separate rule, and we would like each of these rules to be as simple as possible. We encourage this kind of simplicity by updating each element of the network's hidden state, $z_t^i$, using a separate learned update rule defined by a separate set of parameters. Each update rule consists of a multilayer perceptron (MLP) which defines a multiplicative update to be applied to its corresponding latent variable. These "Update MLPs" have access to the values of all elements of the previous timestep's hidden state, $\tilde{z}_{t-1}^i$ (having passed already through its latent bottleneck), as well as to the network's current observations $\mathbf{o}_t$. Each element of the MLP input (i.e. of $\tilde{z}_{t-1}$ and of $\mathbf{o}_t$) must pass through an additional information bottleneck, which we refer to as "Update MLP Bottlenecks". The output of each update MLP is a scalar weight $w_t^i$ and update target $u_t^i$, which are used together to update the value of the associated latent variable $z^i$ in a manner analogous to the Gated Recurrent Unit [9].

$$w_t^i, u_t^i = \text{MLP}_{\text{update}}^i(\text{bottleneck}(\tilde{\mathbf{z}}_{t-1}, \mathbf{o}_t)) \tag{4}$$

$$z_t^i = (1 - w_t^i)\tilde{z}_{t-1}^i + w_t^i u_t^i \tag{5}$$

The latent variables are used on each timestep to make a prediction about the target, $y$, using a separate "Choice MLP":

$$\hat{y}_t = \text{MLP}_{\text{output}}(\tilde{\mathbf{z}}_t) \tag{6}$$

We train the model end-to-end by gradient descent to minimize the total loss function:

$$L_{\text{total}} = L_{\text{softmax}}(y, \hat{y}) + \beta \sum_{b \in \text{bottlenecks}} L_{\text{bottleneck}}(b) \tag{7}$$

where $y$ are supervised targets, $L_{\text{softmax}}$ is a softmax cross-entropy loss. The hyperparameter $\beta$ scales the cost associated with passing information through the bottleneck [23, 8]. Different values of $\beta$ are expected to produce solutions which adopt different tradeoffs between predictive accuracy and model simplicity.

### 3.2 Training Details

For the two-armed bandit task, datasets consisted of sequences of binary choices made (left vs. right) and outcomes experienced (reward vs. no reward) by either a rat [37] or an artificial agent (Q-Learning or Leaky Actor-Critic, see below). Each episode corresponded to a single behavioral session. On each timestep, the observation given to the network consisted of the choice and reward from the corresponding trial, and the target was the choice on the subsequent trial. For the pulse accumulation task, datasets consisted of sequences of integer pulse counts (left vs. right) observed and binary choices made (left vs right) by either a rat [7] or an artificial agent (Bounded Accumulation, see below). Each episode corresponded to a single trial. On each timestep, the observation given to the network consisted of the left and right pulse counts in a corresponding 10ms timebin. The target was null for all timesteps during the stimulus, and on the final timestep of each trial indicated the binary choice made by the rat.

All networks trained for this paper were of the same size and trained in the same way. Each network had five latent variables. Update MLPs consisted of three hidden layers containing five units each. The Choice MLP consisted of two hidden layers of two units each. We used the rectified linear (ReLU) activation function. Networks were defined using custom modules written using Jax [6] and Haiku [22]. Network parameters were optimized using gradient descent and the Adam optimizer [30], with a learning rate of $5 \times 10^{-3}$. We typically trained networks for $10^5$ steps, except that networks with very low $\beta$ ($10^{-3}$ or $3 \times 10^{-4}$) required longer to converge and were trained for $5 \times 10^5$ steps. Using a second-generation TPU, models required between four and fifty hours to complete this number of training steps.

# 4 DisRNN Recovers True Structure in Synthetic Datasets

We first apply DisRNN to synthetic datasets in which the process generating the data is known. We consider a task that has been the subject of intensive cognitive modeling efforts in psychology and neuroscience, the dynamic two-armed bandit task (Figure 2) [13, 28, 26, 43, 37]. In each trial of this task, the agent selects one of two available actions and then experiences a probabilistic reward. In our instantiation of the task, rewards are binary, and the reward probability conditional on selecting each arm drifts independently over time according to a bounded random walk [37].

We generated synthetic datasets from two reinforcement learning agents performing this task: Q-Learning and Leaky Actor-Critic. These agents have markedly different latent variables and update rules, allowing us to test whether DisRNN can recover the correct structure of each agent.

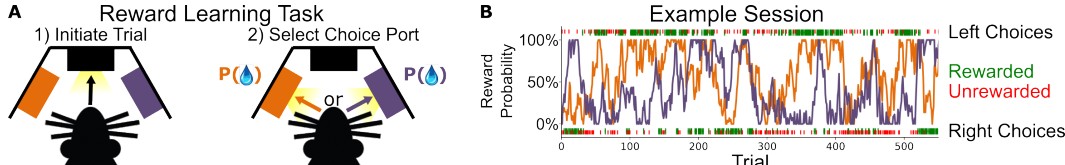

Figure 2: **Dynamic Two-Armed Bandit Task**. **a**: In each trial, the rat selects one of two actions (left or right) and receives one of two outcomes (reward or no reward). **b**: Example behavioral session [37]. Orange and purple lines show the drifting generative reward probabilities for each action. The placement of each tick mark (above or below) indicates the choice of the animal on that trial, while the color of the tick indicates whether or not the animal received reward. We will use this session as a running example to illustrate the dynamics of artificial agents and RNNs.

## 4.1 Q-Learning Agent

The Q-learning agent [48] maintains two latent variables, $Q_{\text{left}}$ and $Q_{\text{right}}$. Each of these is associated with one of the available actions, and gives a running average of recent rewards experienced after taking that action. They are updated according to the following rules:

$$Q_{t+1}(a_t) = (1 - \alpha)Q_t(a_t) + \alpha r_t \tag{8}$$

$$Q_{t+1}(a \neq a_t) = Q_t(a \neq a_t) \tag{9}$$

where $\alpha$ is a learning rate parameter. We visualize these update rules by plotting the updated value of $Q$ as a function of its initial value and of the trial type (Figure 3a). A key feature is that each $Q$ value only changes when its corresponding action is selected.

The Q-Learning agent selects actions based on the difference in $Q$-values:

$$a_t \sim \text{Logistic}\big(\beta(Q_{t,\text{left}} - Q_{t,\text{right}})\big) \tag{10}$$

where $\beta$ is an inverse temperature parameter. We generated a large synthetic dataset consisting of choices made and rewards received by the Q-Learning agent ($\alpha = 0.3$, $\beta = 3$; 1000 sessions of 500 trials each) (Figure 3b). We then trained a DisRNN to imitate this dataset. Inspecting the learned bottleneck parameters of this network (Figure 3e), we find that just two of its latents have open bottlenecks ($\sigma \ll 1$), allowing them to pass information forward through time. We find that the update rules for these latents have open bottlenecks for the input from both previous choice and previous reward. We find that the three latents with closed ($\sigma \approx 1$) bottlenecks take on near-zero values on all timesteps, while the two with open bottlenecks follow timecourses that are strikingly similar to those of $Q_{\text{left}}$ and $Q_{\text{right}}$ (albeit centered on zero instead of on 0.5; Figure 3c). We visualize the learned update rule associated with each latent by probing its associated update MLP (Figure 3d). We find that these update rules are strikingly similar to those for $Q_{\text{left}}$ and $Q_{\text{right}}$. They reproduce the key feature that each is only updated following choices to a particular action.

## 4.2 Leaky Actor-Critic Agent

The Leaky Actor-Critic agent uses a modification of the policy gradient learning rule [48](Section 2.8). Like the Q-Learning agent, it makes use of two latent variables, but uses very different rules

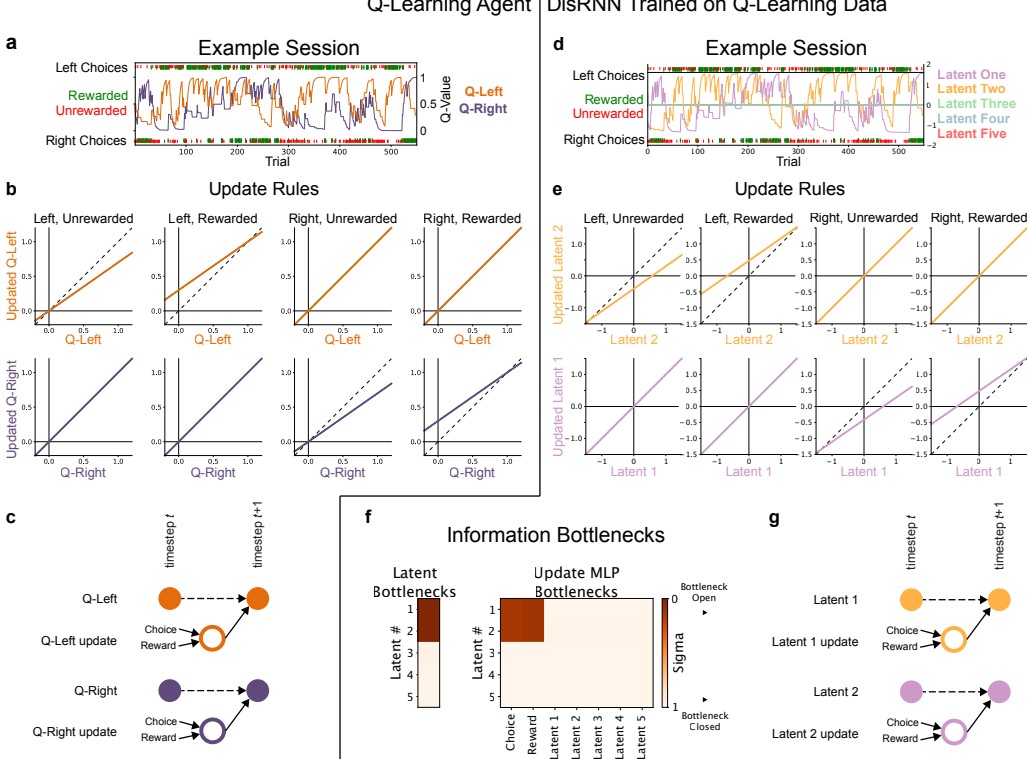

Figure 3: **DisRNN Recovers Latent Dynamics of Q-Learning**. **a**: Q-Learning agent run using the choices and rewards from the example behavioral session (Figure 2) to generate timeseries for $Q_{\text{left}}$ and $Q_{\text{right}}$. **b**: Visualization of the Q-Learning update rules. Each panel visualizes the update of either $Q_{\text{left}}$ (top row) or $Q_{\text{right}}$ (bottom row) following a particular combination of choice and outcome (columns). The post-update value is shown on the vertical axis, as a function of the pre-update value, on the horizontal. Dashed lines are identity. **c**: Dependency graph of Q-learning. Q-learning computes an update for each latent ($Q_{\text{left}}$ and $Q_{\text{right}}$) on each timestep. The update for $Q_{\text{left}}$ depends on choice and reward (Equation 8). The new value of $Q_{\text{left}}$ is a weighted sum of its old value (dashed line) and the update. $Q_{\text{right}}$ is similarly updated. **d**: DisRNN trained on a synthetic behavioral dataset generated by the Q-Learning agent. This panel shows data from the trained model run with choice and reward input from the same example session. Latents have been ordered, signed, and colored to highlight similarities with the Q-Learning agent. **e**: Visualization of the learned update rules (equations 4 and 5) for the first two latents. **f**: Learned parameters of the information bottlenecks. Left: Transmission bottlenecks (equation 3). Right: Update MLP bottlenecks (equation 4). Darker colors indicate open bottlenecks: for example, the update for latent 1 can depend on choice and reward. **g**: Dependency graph of cognitive model learned by DisRNN.

to update them. The first variable, $V$, tracks "value". It keeps a running average of recent rewards, regardless of the choice that preceded them:

$$V_{t+1} = (1 - \alpha_v)V_t + \alpha_v r_t \tag{11}$$

where $\alpha_v$ is a learning rate parameter.

The second, $\Theta$, is a "policy" variable that determines the probability with which the agent will select each action. The policy gradient algorithm defines two variables $\theta_{left}$ and $\theta_{right}$, each associated with one of the actions and updated using the following rules:

$$\theta_{t+1}(a_t) = \big(1 - \alpha_f\big)\theta_t(a_t) + \alpha_l\big(r_t - V_t\big)\big(1 - \pi_t(a_t)\big) \tag{12}$$

$$\theta_{t+1}(a \neq a_t) = (1 - \alpha_f)\theta_t(a) - \alpha_l\big(r_t - V_t\big)\pi_t(a) \tag{13}$$

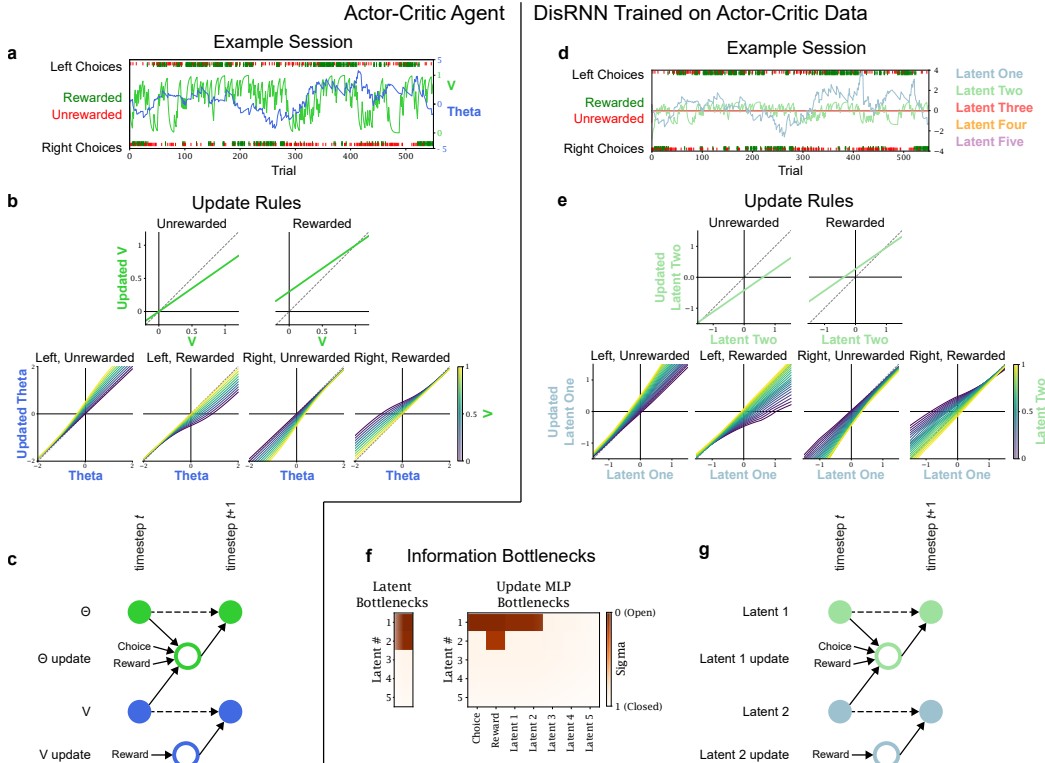

Figure 4: **DisRNN Recovers Latent Dynamics of Leaky Actor-Critic**. **a**: Actor-Critic agent run on the example rat behavioral session (Figure 2) to generate timeseries for $V$ and $\Theta$. **b**: Visualization of the Actor-Critic update equations. The update for $v$ depends on reward but is independent of choice. The update for $\Theta$ depends on choice and reward, as well as the value of $V$. **c**: Dependency graph of Actor-Critic learning. **d**: DisRNN trained on a synthetic behavioral dataset generated by an Actor-Critic agent. **e**: Visualization of the learned update rules for the first two latents. **f**: Learned bottleneck parameters. Left: Transmission bottlenecks (equation 3). The first two latents have open bottlenecks; the remaining three latents have closed bottlenecks. Right: Update MLP bottlenecks (equation 4). **g**: Dependency graph of DisRNN trained on Actor-Critic data.

where $\pi_t$ is the agent's policy defined below (equation 14), $\alpha_l$ is a learning rate parameter, and $\alpha_f$ is a forgetting rate parameter that we have added to the update rule. This forgetting mechanism causes the policy variables to decay towards zero over time, allowing the agent to perform in an environment with changing reward probabilities. In a setting with just two available actions, $\theta_{\text{left}}$ and $\theta_{\text{right}}$ are degenerate: it will always be the case that $\theta_{\text{left}} = -\theta_{\text{right}}$. We therefore define a single latent variable $\Theta = \theta_{\text{left}} - \theta_{\text{right}}$ for the purposes of visualization. Actions are selected according to:

$$a_t \sim \text{Logistic}(\Theta) \tag{14}$$

We visualize these update rules by plotting the updated values of $V$ and of $\Theta$ as a function of their initial value and of the trial type (Figure 4b). These update rules are quite different from those of the Q-Learning agent. One key feature is that the update rule for $V$ depends only on reward, not on choice. Another is that the update rule for $\Theta$ depends not only on choice and reward, but also on $V$.

We generated a large synthetic dataset consisting of choices made and rewards received by the Leaky Actor-Critic agent ($\alpha_V = 0.3$, $\alpha_L = 1$, $\alpha_F = 0.05$; 1000 sessions of 500 trials each), and trained a DisRNN to imitate this dataset. Inspecting the learned bottleneck parameters of this network (Figure 4e), we find that two of its latents have open bottlenecks allowing them to pass information forward through time. One of these (latent two) has an open bottleneck to receive input from reward only. We find that this latent is strikingly similar to $V$ from the Leaky Actor-Critic agent, both in terms

of its timecourse (Figure 4c) and update rule (Figure 4d, above). The other (latent one) has open bottlenecks to receive input from choice, reward, and from latent two. This latent is strikingly similar to Θ from the Leaky Actor-Critic Agent, both in terms of its timecourse (Figure 4c) and update rule (Figure 4d, below).

Together with the previous section, these results indicate that DisRNN is capable of recovering the true generative structure of agents with two very different cognitive strategies to perform the two-armed bandit task. We also applied DisRNN to a synthetic behavioral dataset from an artificial performing a very different behavioral neuroscience task: decision-making by accumulation of noisy evidence (Appendix A). We found that it was similarly able to recover the true generative structure of this agent (Figure A1)

## 5 DisRNN Reveals Interpretable Models of Rat Behavior

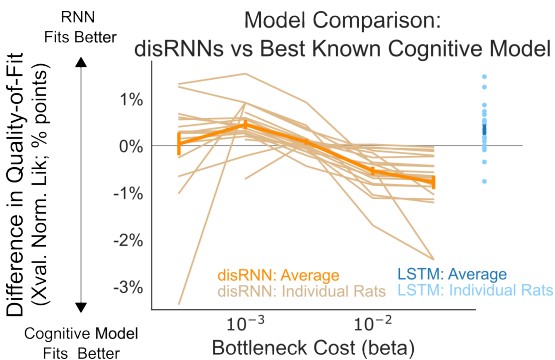

Figure 5: **Model Comparison on Rat Behavioral Dataset**. Quality of model fit (cross-validated normalized likelihood) relative to the human-derived cognitive model from [37] for DisRNN (orange) and LSTM (blue). The x-axis for DisRNN represents different values of the hyperparameter $\beta$, which controls the tradeoff between model simplicity and predictive performance. LSTM hyperparameters were selected by subject-level cross-validation. Error bars indicate standard error over $N = 20$ rats.

Having established that DisRNN can recover true cognitive mechanisms in synthetic datasets with known ground truth, we moved on to consider a large laboratory dataset from rats performing the drifting two-armed bandit task [37]. This dataset has previously been the subject of an intensive human effort at data-driven cognitive modeling, which resulted in a cognitive model consisting of three components each with its own latent variable: a fast-timescale reward-seeking component, a slower perseverative component, and a very slow "gambler's fallacy" component. This model provided a better fit to the dataset than existing models from the literature, and is currently the best known cognitive model for this dataset [37]. We first asked whether DisRNN could provide a similar quality of fit to this model. We fit DisRNNs using four different values of the parameter $\beta$ ($10^{-3}$, $3\text{x}10^{-3}$, $10^{-2}$, and $3\text{x}10^{-2}$) that controls the relative weight of predictive power and model simplicity in the network's loss function (equation 7). Following [37], we evaluated model performance using two-fold cross-validation: we divide each rat's dataset into even-numbered and odd-numbered sessions, fit a set of model parameters to each, and compute the log-likelihood for each parameter set using the unseen portion of the dataset. To compare results across different animals, we use "normalized likelihood": $e^{\text{log-likelihood/n trials}}$ [12]. For each rat, we compute this both for our DisRNNs and for the cognitive model from [37], and plot the differences in Figure 5. We find that DisRNNs trained with $\beta = 3\text{x}10^{-3}$ achieved a quality of fit similar to the cognitive model, and that those with $\beta = 10^{-3}$ even narrowly outperformed it (Figure 5, orange curve). As an additional benchmark, we also compared our models to a widely-used RNN architecture, the LSTM [25]. We selected LSTM hyperparameters (network size and early stopping) using subject-level cross-validation: we divided the dataset into subsets containing only even-numbered and odd-numbered rats, identified the hyperparameters that maximized (session-level) cross-validated likelihood in each subset, and evaluated networks with those hyperparameters in the unseen subset. We found that DisRNNs with $\beta = 10^{-3}$ achieved a very similar quality of fit to these optimized LSTMs. Together these results indicate the DisRNNs, despite their architectural constraints and disentanglement loss,

can achieve predictive performance comparable to standard RNNs and to well-fit human-derived cognitive models.

Finally, we examined the parameters of DisRNNs fit to the complete dataset for each rat. We found that these were typically low-dimensional, with only a small number of latent variables having open bottlenecks, as well as sparse, with each update MLP having only a small number of open bottlenecks. While a thorough characterization of the patterns present in DisRNNs fit to individual rats remains a direction for future work, we present a representative example in Figure 6. This figure shows three DisRNNs with different values of $\beta$ fit to the dataset from the same example rat. The DisRNN with the largest $\beta$ of $3\text{x}10^{-2}$ (left) adopts a strategy that utilizes only one latent variable. This strategy mixes reward-seeking and perseverative patterns into a single update rule (note that fixed points, where the line crosses unity, are more extreme for the rewarded conditions than the unrewarded ones). The DisRNN with a medium $\beta$ of $10^{-2}$ (middle) adds a second latent variable that is purely perseverative and operates on a considerably slower timescale. Its fast-timescale reward-seeking latent variable, however, shows a similar perseverative signature to the previous network. The network with the smallest $\beta$ of $3\text{x}10^{-3}$ (right) retains both the fast reward-seeking and slower perseverative dynamics and adds a third, much slower-timescale, latent. The update rule for this latent depends both on reward and on the value of the perseverative latent. Interestingly, this very slow latent influences the update rules for both the reward-seeking and the perseverative latents. While the best human-derived cognitive model does include a very long-timescale component, this component simply influences choice rather than modulating the update rules of the other components. Situations where one latent variable modulates the update rule of another do sometimes occur in human-derived models for related tasks [3, 42], though it is not clear whether the pattern seen in this network is consistent with any of these. Characterizing these patterns in more detail, and especially quantifying similarities and differences among different animals, will be an important step for future work.

We also applied DisRNN to a laboratory behavioral dataset from rats performing an evidence accumulation task [7], and found that it was similarly able to identify low-dimensional models with interpretable structure (Figure A2).

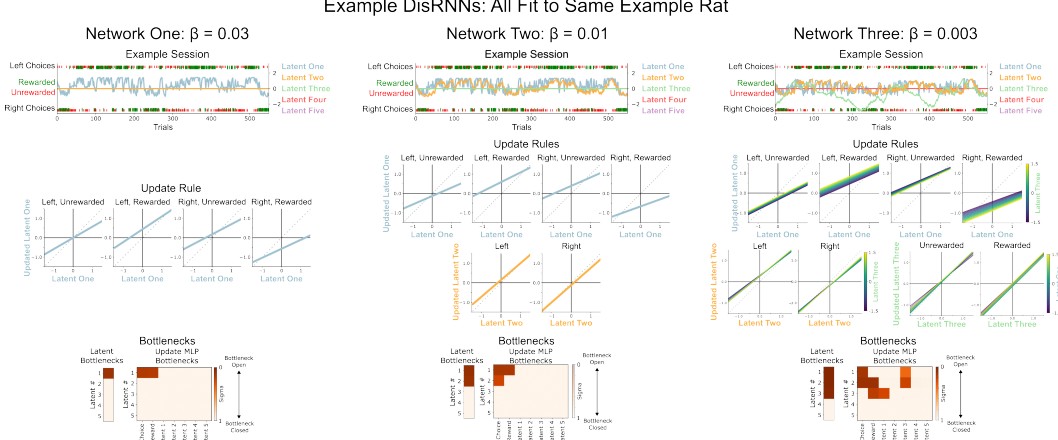

Figure 6: **DisRNN Trained on Rat Datasets**. Examples of fit DisRNN networks with different values of the hyperparameter $\beta$ controlling the tradeoff between simplicity and predictive power.

# 6  Discussion

In this work, we develop a framework for discovering parsimonious cognitive models directly from behavioral data by fitting recurrent neural networks that contain structural features that encourage them to learn sparse, disentangled representations. Fit to synthetic datasets generated using known cognitive mechanisms, our method accurately recovers the structure of those mechanisms. Fit to laboratory datasets from rats performing a cognitive task, our method reveals models that are relatively simple and human-interpretable while outperforming the best known human-derived cognitive model in terms of predictive accuracy.

While the extent to which a model fit is "human-interpretable" is of course ultimately subjective, we believe that sparsity and disentangling provide benefits for at least three distinct reasons. The first is that fully understanding a model requires a human expert to inspect the update rules. The smaller the number of latents and fewer inputs to the update rule for each, the less cognitive burden will be placed on that expert, and the more likely they will be able to arrive at a satisfying human intuition about the cognitive mechanism embodied by the model. The second is that goal of discovery is to identify models that human experts will consider to be cognitively plausible. When evaluating classic handcrafted models, many experts agree that, all else being equal, simpler models (smaller number of equations, fewer terms in each equation) are more plausible. The third is that such a model is more likely to be useful for scientific tasks, such as searching for correlates in measurements of neural activity, that involve interacting with finite datasets.

The fit disRNNs have several key features than enable them to be applied immediately to standard cognitive neuroscience workflows [51, 12, 10, 39]. The first is that they generate timestep-by-timestep timecourses for the values of latent variables that play known cognitive roles within the model. These timecourses can be used as predictions about neural activity: if the model's mechanisms are implemented in the brain, then somewhere in the brain there is likely to be a signal that follows a similar timecourse. A second key feature is that disRNN makes explicit the rules by which latent variables are updated. These can be used as predictions about update rules within the brain: if the model's mechanisms are implemented in the brain, then somewhere in the brain there are likely to be similar update rules (e.g. synaptic weights to update the ongoing neural activity, synaptic plasticity to adjust synaptic weights, etc). A third is that a trained disRNN can be used to generate predictions for experiments which alter neural activity, for example by silencing it using optogenetics. Activity within the disRNN can be altered, for example by artificially zero-ing out a particular latent on a particular subset of trials, and the model can be run to generate predictions for behavioral and neural data [38].

An appealing feature of disRNN is that, for the same dataset, models with different complexity can be discovered by training networks with different values of the hyperparameter $\beta$. For some applications, it may be best to select the model that achieves the highest cross-validated quality-of-fit. For others, it may be useful to work with a model that is simpler, even though it may achieve lower quality-of-fit. We expect that in practice disRNN will be most useful if researchers fit several copies, at each of several values of $\beta$, and inspect their fits to determine which (if any) have learned models that are useful (Appendix B).

The most important limitation of our approach is that there is no guarantee that the recovered model will necessarily correspond to the true cognitive mechanisms used by the brain. Instead, it uses the dataset to reveal a parsimonious hypothesis. This hypothesis that requires evaluation by a human scientist to determine whether, given the rest of what is known about the psychology and biology of the system, it is plausible. If plausible, it may require new experiments to test the predictions that it makes. This limitation is fundamental to any approach that seeks to make inferences about cognitive mechanisms by analyzing behavioral data, as any behavioral dataset will provide only a limited window on neural mechanisms. One view of these systems is that they perform automatic hypothesis generation, without themselves addressing the problem of hypothesis testing.

An additional limitation is that the method, because it requires fitting quite flexible models, likely is applicable only to relatively large-scale datasets. Determining how performance scales with dataset size and exploring methods of improving data efficiency may be important directions for further research. Another direction is exploring the performance of disRNN on behavioral datasets from other types of cognitive neuroscience tasks. A large number of scientifically-important tasks exist which are of similar complexity to those considered here, and we believe that for many of these disRNN would prove able to discover reasonable models. A large number of tasks also exist which are considerably more complex, and we expect that for these disRNN in its current form might struggle, perhaps by requiring prohibitively large datasets.

The ultimate goal in cognitive model discovery is to develop tools which can reveal scientifically valuable models automatically from data. We believe that disRNN represents an important step towards this goal. Its most important test will come from work applying it to novel datasets, for which no model is known, and evaluating whether the structures it learns are useful to researchers.

## Acknowledgments and Disclosure of Funding

We would like to thank Kim Stachenfeld and Yu Jin Oh for helpful comments on the manuscript.

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

# A Pulse Accumulation Task

In this appendix, we explore the generality of disRNN by applying it to datasets from a very different class of tasks: decision-making via accumulation of noisy evidence. Like the reward-learning tasks considered earlier, these tasks are heavily studied in behavioral neuroscience and have been the subject of intensive cognitive modeling efforts. We consider specifically the "pulse accumulation" tasks, which are particularly amenable to computational modeling [7, 45, 14, 41]. In each trial of these tasks, discrete "pulses" of evidence arrive over time via two parallel streams (typically "left" and "right"), and the agent is rewarded for reporting which of the two streams contained a larger total number of pulses.

## A.1 Bounded Accumulator Agent

We generated synthetic data from an artificial decision-making agent performing this task by implementing a known strategy: Bounded Accumulation. The Bounded Accumulator agent maintains a single latent variable $A$, which indicates a running estimate of the difference in pulse counts between the two streams. At the beginning of each trial, $A$ is initialized to $0$ and then updated timestep-by-timestep as the pulses arrive according to:

$$A_{t+1} = A_t + \sigma_a \eta_a + \delta_{t,t_r} \sigma_p \eta_r + \delta_{t,t_l} \sigma_p \eta_l \tag{15}$$

where $\sigma_a$ is an accumulator noise parameter controlling noise that scales with time; $\sigma_c$ is a pulse noise parameter controlling how much sensory noise is associated with each evidence pulse; $\delta_{t,t_{R,L}}$ are delta functions are the times of the right and left pulses; $\eta_{R,L}$ are i.i.d. Gaussian variables drawn from $\mathcal{N}(1,1)$; $\eta_a$ is an i.i.d. Gaussian variable drawn from $\mathcal{N}(0,1)$. The dynamics of $A$ are additionally governed by a bound parameter $B$. If $A$ reaches a value more extreme than $\pm B$, it will remain at this value for the remainder of the trial.

The agent's decision on each trial is based on the final value of $A$ and on a lapse parameter $l$, which is constrained to lie between 0 and 0.5. With probability $1 - l$, the decision is given by the sign of $A$, and with probability $l$ it is inverted.

We generated a large synthetic dataset consisting of pulses observed and decisions made, using a Bounded Accumulator agent with relatively little noise ($\sigma_a = 0$; $\sigma_p = 0.01$; $B$=2.9; $l$=0; 200,000 trials) (Figure A1a,b). We then trained a DisRNN to imitate this dataset. We find that just one of its latents has an open bottleneck, allowing it to pass information through time, and that the timecourse of this latent is strikingly similar to that of $A$ (Figure A1c). We visualize the learned update rules for this latent by probing its associated update MLP, and find that they are strikingly similar to those for $A$ (Figure A1d).

## A.2 Rat Behavioral Data

We next considered a large laboratory dataset from rats performing a pulse accumulation task in which the evidence streams consisted of auditory clicks delivered from a pair of speakers, one located to the left and one located to the right of the rat [7]. For the purposes of modeling, we discretized each trial into 10ms timebins, and gave as inputs to the network the integer number of clicks that were delivered from each speaker in each bin. We fit copies of DisRNN to the dataset from each individual rat, with different values of the bottleneck cost $\beta$.

We find that these often recover human-interpretable models which capture known features of rat behavior on this task. In the first example shown (Figure A2, left), the model has identified a single latent (green) which follows a timecourse similar to that of an accumulator variable. Unlike in the synthetic agent considered above, this accumulator does not seem to have a bound, and it does seem to have "decay" dynamics, drifting gradually towards 0 on timesteps that do not contain clicks. In the second example shown (Figure A2, left), the model has identified a similar accumulator latent (green), whose dynamics are now modified by those of a second latent (blue). The value of this second latent decreases sharply in response to clicks (from either speaker) and recovers gradually during timesteps without clicks. While it remains low, the impact of additional clicks on the accumulator variable is attenuated. This pattern is consistent with "sensory adaptation" dynamics that are often included in cognitive models for tasks of this kind [7].

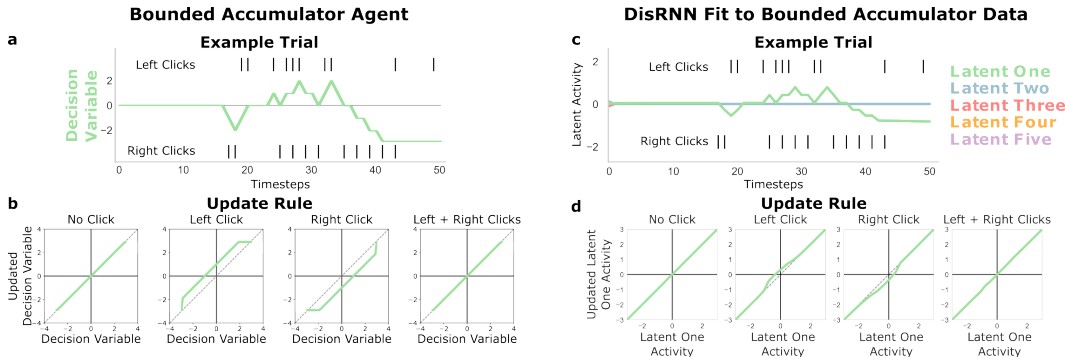

Figure A1: **DisRNN Recovers Latent Dynamics of Bounded Accumulator**. **a**: Bounded Accumulator agent run on an example trial of the click accumulation task to generate timeseries for the accumulator variable $A$. **b**: Visualization of the Bounded Accumulator update equation. $A$ is incremented on timesteps containing a left click only and decremented on trials containing a right click only, unless it has already reached one of the bounds (here $\pm3$), in which case it remains at that bound. On timesteps with no click or with clicks on both sides, $A$ remains the same. The dynamics of $A$ beyond the bound are undefined, as these values are never reached **c**: DisRNN trained on a synthetic behavioral dataset generated by an Bounded Accumulator agent. **d**: Visualization of the learned update rules for the first latent.

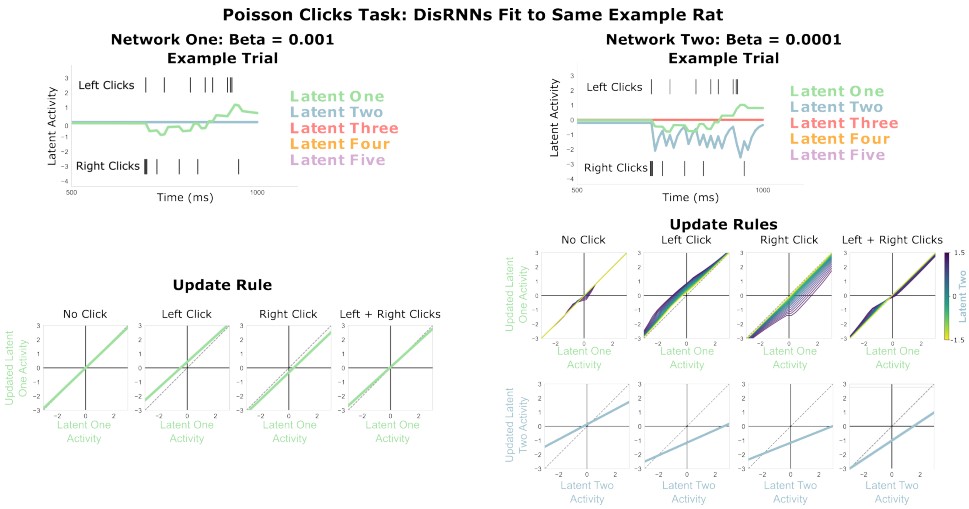

Figure A2: **DisRNN Trained on Rat Decision-Making Dataset**. Examples of fit DisRNN networks with different values of the hyperparameter $\beta$ controlling the tradeoff between simplicity and predictive power

# B    Robustness of Fits

In this appendix, we explore the robustness of disRNN fits to different values of the hyperparameter $\beta$, which controls the contribution of the information bottleneck cost to the loss function (equation 7). We fit three copies of DisRNN to each of our three synthetic datasets (Q-Learning, Actor-Critic, and Bounded Accumulation) at each of nine values of $\beta$ ranging from $10^{-5}$ to $10^{-1}$ (Figure A3). For each DisRNN, we measure the cross-validated quality-of-fit (Figure A3, yellow, top row), as well as number of information bottlenecks that are open ($\sigma < 0.3$) for both latent bottlenecks (blue, middle row) and update bottlenecks (green, bottom row). We compare the number of open latent bottlenecks in the fit DisRNNs to the number of latent variables in the true generative process (Q-Learning and Actor-Critic: two latent variables; bounded Accumulator: One latent variable), and the number of open update bottlenecks to the number of total terms across all update rules in the true generative processes (Q-Learning: four terms; Actor-Critic: five terms; Bounded Accumulator: three terms).

We find for all three datasets that DisRNNs with very high values of $\beta$ adopted structures simpler than those of the true generative agents (fewer open bottlenecks), and that these DisRNNs earned lower quality-of-fit. We find for all three datasets that DisRNNs with very low values of $\beta$ adopted structures more complex than those of the true generative agents (more open bottlenecks), and that these typically earned a high quality-of-fit. In between the extremes, DisRNNs with medium values of $\beta$ adopted structures similar to those of the true generative agents. These DisRNNs could reliably be identified as those with the simplest structures (largest $\beta$; fewest open bottlenecks) among the set of networks with high quality-of-fit. Manual inspection revealed that all DisRNNs with the correct number of open bottlenecks also adopted the correct model structure and qualitatively well-matched update rules (data not shown).

These results demonstrate that an approach of fitting multiple copies of DisRNN with different hyperparameters is able to reliably identify the true model structure in each of our synthetic datasets. They also highlight the fact that this approach results in a spectrum of models adopting different tradeoffs between simplicity and quality-of-fit. We expect that DisRNN will be most useful in practice if researchers fit several models with different values of $\beta$ and inspect each of them to determine which (if any) have discovered structures of scientific interest.

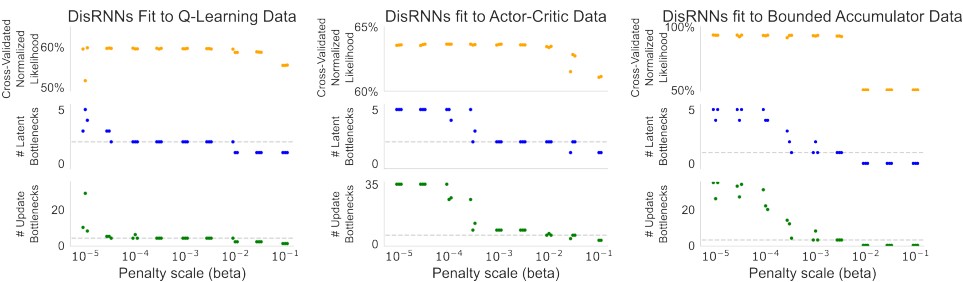

Figure A3: **Robustness of DisRNN Fits Across Different Values of** $\beta$. See text for description

## C  Cognitive Model Discovery Using Classic RNNs

In this appendix, we explore whether similar results might have been obtained using classic neural network architectures, either by constraining the size of the network [2] or by fitting a larger network and considering only the top few principal components of its activity space. We consider networks using the popular Gated Recurrent Unit (GRU) architecture [9], because related work has reported good performance with this architecture on similar datasets [2, 16] and because having only a single type of recurrent unit makes them simpler to analyze.

First, we explore quality of fit for GRU networks of different sizes, ranging from one to fourteen hidden units, fit to each of the five datasets (table 1). We find that for the synthetic datasets, "tiny" networks with just one or two recurrent units do provide a quality-of-fit that is only slightly less than that of larger networks. This raises the possibility that these networks might be able to discover the true model structure for these synthetic datasets. For the rat datasets, we find that larger networks provide a meaningfully better quality-of-fit. This indicates that very small networks do not quantitatively fully capture the structure present in the rat datasets, though it does not rule out the possibility that they might nevertheless discover scientifically-useful model structures.

Next, we inspected the fit of the "tiny" two-unit GRU networks to our three synthetic datasets, examining plots of their update rules (Figure A4). We see that these typically do not have a 1:1 relationship with the true generative latent variables. The exception is the Q-Learning agent, for which two-unit GRUs do often discover a disentangled solution. Solutions found for the Actor-Critic and Bounded Accumulator datasets are fully entangled, with each unit's update dependent on the value of the other unit and on both input variables. We interpret this to mean that very small conventional networks can discover dynamics that recapture those of certain generative processes, but that they do not do so reliably.

Finally, we inspected the fit of larger ten-unit GRU networks to our three synthetic datasets (Figure A5). We summarize their dynamics by plotting update rules for the first two principal components. While some interpretable patterns are apparent, there is still not a 1:1 mapping between PCs and

the latent variables of the generative process. The dynamics are entangled, with each PC's update depending on all inputs and on the value of the other PC.

Taken together, we interpret these results to indicate that conventional neural networks like GRUs can be a viable route to cognitive model discovery in some circumstances, but also that they have important limitations. One limitation is that, while task training ensures that the dynamics they contain are sufficient to solve the task, nothing ensures that all aspects of these dynamics are necessary (they are free to retain null-space dynamics). Another limitation is that, while the number of latent variables can be constrained by limiting network size or by only considering the top few PCs, nothing ensures that the update rules for these variables are sparse, and nothing encourages them to be "axis aligned", mapping 1:1 onto the true generative dynamics. Note that the "tiny RNNs" approach of Ji-An et al., [2] introduced additional architectural changes to the networks, which may help address some of these limitations.

| Hidden | Q-Learning | Actor-Critic | Bandit Rats | Accumulator | Clicks Rats |
|--------|-----------|--------------|-------------|-------------|-------------|
| 1 | -0.72 | -1.38 | -2.16 | - | -0.36 |
| 2 | - | - | - | -0.008 | - |
| 3 | 0.002 | 0.01 | 0.34 | 0.001 | 0.08 |
| 4 | 0.003 | 0.02 | 0.26 | 0.003 | 0.07 |
| 5 | 0.002 | 0.02 | 0.39 | 0.004 | 0.08 |
| 6 | 0.002 | 0.02 | 0.38 | 0.004 | 0.09 |
| 7 | 0.002 | 0.02 | 0.35 | 0.004 | 0.08 |
| 8 | 0.002 | 0.02 | 0.46 | 0.004 | 0.10 |
| 9 | 0.002 | 0.02 | 0.44 | 0.004 | 0.11 |
| 10 | 0.002 | 0.02 | 0.38 | 0.004 | 0.05 |
| 11 | 0.002 | 0.02 | 0.35 | 0.004 | 0.10 |
| 12 | 0.002 | 0.02 | 0.51 | 0.004 | 0.10 |
| 13 | 0.001 | 0.02 | 0.53 | 0.004 | 0.10 |
| 14 | 0.001 | 0.02 | 0.50 | 0.004 | 0.13 |

Table 1: **Model Comparison: GRUs of Different Sizes** Difference in cross-validated normalized likelihood (percentage points) between GRU networks with different numbers of hidden units. Score for each synthetic dataset is the average of three networks fit with different random seeds. Score for the rat two-armed bandit dataset is the average of three networks for each of 20 rats. Score for the rat Poisson clicks dataset is the average of three networks for each of 19 rats. Scores are reported as the difference between the average score for that combination of dataset and network size and a reference size for each dataset (two units for Q-Learning, Actor-Critic, Bandit Rats, and Clicks Rats; one unit for Bounded Accumulator)

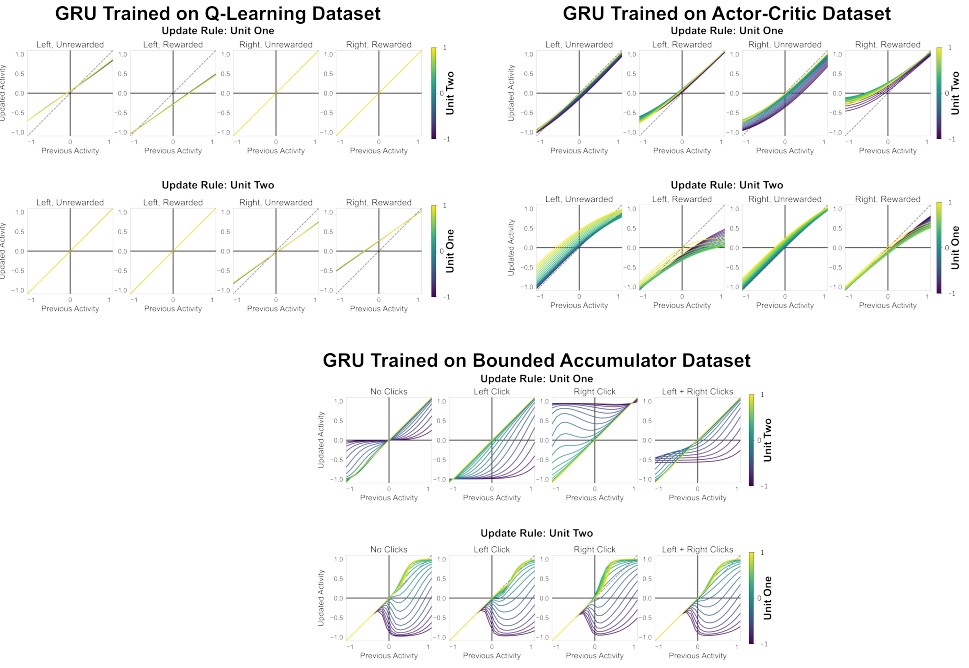

Figure A4: **Update Rules for Two-Unit GRUs Fit to Synthetic Datasets**. The update rules for the GRU fit to the Q-Learning dataset are strikingly similar to the true update rules of the Q-Learning agent (Figure 3). The update rules for GRUs fit to the Actor-Critic and Bounded Accumulator datasets are quite different from the true update rules of these agents.

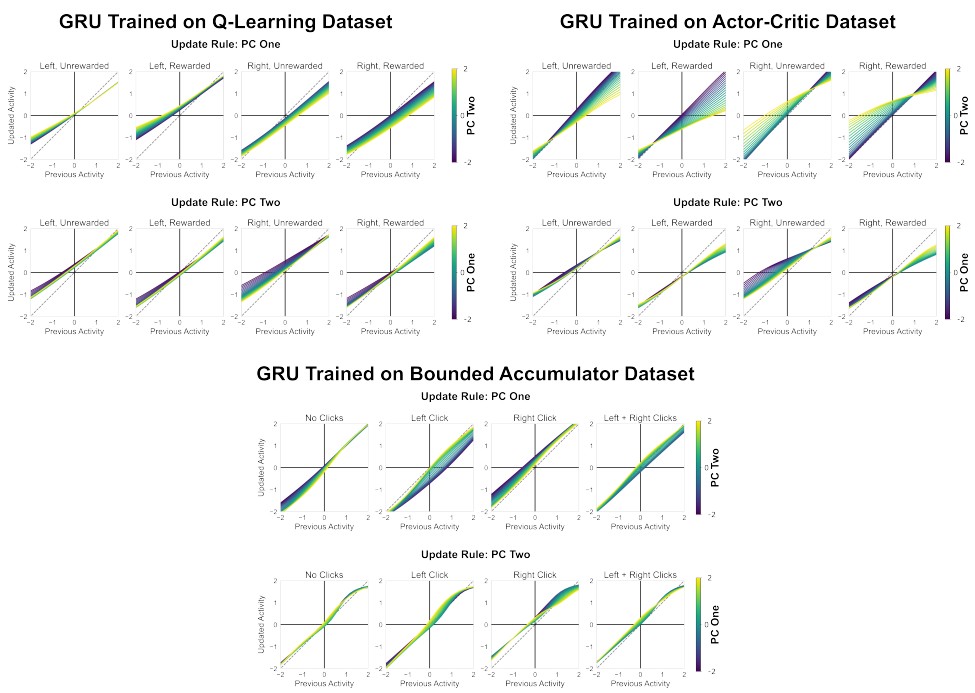

Figure A5: **Update Rules for Top Two PCs of Large GRUs fit to Synthetic Datasets**. For all three networks, the update rules for the top two PCs are quite different from the true update rules of the corresponding generative agents.

