# OpenReview forum: "Cognitive Model Discovery via Disentangled RNNs"
_NeurIPS.cc/2023/Conference — NeurIPS 2023 poster_

### Official Review · Reviewer_u93Y · 2023-07-05

**Soundness:** 2 fair
**Presentation:** 3 good
**Contribution:** 2 fair
**Rating:** 5
**Confidence:** 4

**Summary:**

This paper presents an RNN with bottlenecks (both a sampling and a transition bottleneck) and applies the RNN to behavioural data from simple RL models, as well as rodent data. It is shown that the model learns interpretable latents, and ‘rediscovers’ q-learning as well as actor-critic learning and is able to account for the rodent data as well as the current best cognitive models.

**Strengths:**

This direction is interesting, and relevant to the animal behaviour and neuroscience literature. The technical details are sound.

**Weaknesses:**

1)	The model is bespoke – a separate MLP for each element of z, as well as the bottleneck. It would be helpful to have an empirical understanding of how important these elements are to learn a disentangled RNN (e.g. via ablation).

2)	It would also be helpful to understand the benefit of disentanglement, i.e. if you just trained a standard RNN then looked at the first few principle components do they correspond to the disentangled latents?

3)	The benefit of this approach is that it can offer an understanding of how animals learn. But there is only one analysis of a single rodent experiment, which does not unveil much beyond existing cognitive models.

4)	No analysis of model latent dynamics. Or learning dynamics – does the model go through several stages of understanding? Does this map onto animal behaviour?


**Questions:**

I’m surprised that the models take up to 40 hours to train on a TPU?! The models are pretty small…

See weaknesses for other questions.


**Limitations:**

See weaknesses for limitations.

I'm not sure if this counts as a limitation or not, but I found it hard to place this work with other concurrent-ish preprints that are conceptually similar. E.g. Li et al., 2023 (cited in this work) that came out 1-2 months prior which trains a RNN on cognitive tasks (but without disentanglement) and is interpretable, but is somewhat more complete in terms of model analysis and relationships to animal data.

---

> ### Author Rebuttal · Authors · 2023-08-10
>
> **Weaknesses**
> > The model is bespoke – a separate MLP for each element of z, as well as the bottleneck. It would be helpful to have an empirical understanding of how important these elements are to learn a disentangled RNN (e.g. via ablation).
>
> We will add an ablation analysis as new supplemental information, removing the disentanglement loss for either the latent bottlenecks only or the update rule bottlenecks only.
>
> > It would also be helpful to understand the benefit of disentanglement, i.e. if you just trained a standard RNN then looked at the first few principle components do they correspond to the disentangled latents?
>
> We informally explored this in previous work, and found that examining the dynamics of the first few principal components of activity in an LSTM typically does not reveal disentangled dynamics. One reason for this is that the LSTM typically will retain dynamical modes that are present prior to training, just from its random initialization, because it is under no pressure to unlearn them. Frustration with this approaches is in fact what led us to the disRNN ideas of 1) explicitly penalizing the network for retaining information in its activations that it was not actually using for anything and 2) allowing the update rule for those activations to be free-form, parameterized by a feedforward sub-network, rather than constrained by particular update equations.
>
> We will add new analysis demonstrating this in our synthetic datasets as new supplemental information.
>
> > The benefit of this approach is that it can offer an understanding of how animals learn. But there is only one analysis of a single rodent experiment, which does not unveil much beyond existing cognitive models.
>
> We have added analysis of a new experiment: decision-making via accumulation of evidence (rebuttal pdf second and third row).
>
> > No analysis of model latent dynamics. Or learning dynamics – does the model go through several stages of understanding? Does this map onto animal behaviour?
>
> The question of the model's learning dynamics is an interesting one to explore in future work. We have focused on using disRNN as a tool for discovering the asymptotic dynamics that govern behavior, but it is true that it could itself be considered as a hypothesis about how animals meta-learn these kinds of tasks. This would involve analyzing in detail the trajectories by which disRNN discovers the asymptotic dynamics, and comparing them to animal's learning trajectories. With respect to analysis of latent dynamics of the trained model, this is a central focus of our work: DisRNN facilitates understanding of these dynamics by allowing us to visualize the update rules that govern them.
>
> **Questions**
>
> > I’m surprised that the models take up to 40 hours to train on a TPU?! The models are pretty small…
>
> This surprised us as well! Typically the networks achieve good predictive performance very quickly (minutes to tens of minutes), but then require a much larger number of training steps to identify disentangled solutions. We expect that suitable schedules of learning rate and bottleneck penalty might substantially reduce training time, and plan to explore this in future work.

---

> > ### Comment · Reviewer_u93Y · 2023-08-13
> > **Many thanks for the response**
> >
> > Many thanks for your responses. I appreciate the additional behavioural task. I have raised my score.

---

### Official Review · Reviewer_Zh1M · 2023-07-07

**Soundness:** 3 good
**Presentation:** 3 good
**Contribution:** 2 fair
**Rating:** 6
**Confidence:** 2

**Summary:**

The authors propose a method for discovering cognitive models automatically by fitting them to behavioral data. They use a recurrent neural network and pass its output through a bottleneck (implemented using variational autoencoders) which is expected to extract relevant cognitive variables. They evaluate the model on synthetic data and on behavioral data from rats performing a reward learning task. The dataset consisted of sequences of binary choices made (left vs. right) and outcomes experienced (reward vs. no reward). The authors evaluated agents based on Leaky Actor-Critic and Q-learning. They used supervised learning and trained the networks to `` imitate this dataset''. The network had a total of five bottlenecks, while the task had two latent variables. After training, the authors found that only two bottlenecks were ``open'' and corresponded to the latent variables. This demonstrated the ability of the network to discover the latent variables.

**Strengths:**

This is quite an original effort to automatically learn cognitive models from the data. The use of bottlenecks was reasonable and well-justified. The results are consistent with the hypothesis as the model was able to recover the two latent variables and use only the necessary bottlenecks.

**Weaknesses:**

The model is fitted only to a single task, which makes it difficult to evaluate whether it would scale and how useful it would be for the broader community.

It is not clear how robust the approach is - while in the particular setups, VAEs converged to the expected values, it is not known if that would happen for different hyperarameters (for example, what if the number of bottlenecks was different or if learning rate or some other hyperparameter was chosen differently).

**Questions:**

Can you identify other tasks where this approach could be used?

How should one go about choosing the number of bottlenecks? Should it always be larger than what is expected that the model will need and if yes, how larger?

**Limitations:**

The authors discussed limitations.

---

> ### Author Rebuttal · Authors · 2023-08-10
>
> **Weaknesses**
> > The model is fitted only to a single task, which makes it difficult to evaluate whether it would scale and how useful it would be for the broader community.
>
> We have now added similar results from an additional task: sensory decision-making via accumulation of noisy evidence. In this task, as we showed in the RL task, disRNN successfully recovers the structure of a handcrafted cognitive model, and reveals a plausible human-interpretable model when fit to a large rat dataset. This task is representative of a very heavily-studied class of decision-making tasks that is widely used in behavioral neuroscience because it isolates an important cognitive process thought to be a building block of cognition. The two-armed bandit task is likewise representative of a very heavily-studied class of “reward learning” tasks thought to isolate a different building block. We expect the method to be immediately useful for other tasks from these domains and others of similar complexity, make up a large fraction of behavioral neuroscience research. We agree that exploring how it will scale to tasks of much greater complexity, especially in the face of limited dataset sizes, will be an important question for future research.
>
> > It is not clear how robust the approach is - while in the particular setups, VAEs converged to the expected values, it is not known if that would happen for different hyperarameters (for example, what if the number of bottlenecks was different or if learning rate or some other hyperparameter was chosen differently).
>
> We have added simulations exploring different values of the weighting parameters $\\beta$ (rebuttal pdf, bottom row), which impacts the complexity of the models that are discovered.
>
> **Questions**
> > Can you identify other tasks where this approach could be used?
>
> See response above
>
> > How should one go about choosing the number of bottlenecks? Should it always be larger than what is expected that the model will need and if yes, how larger?
>
> For a network to learn disentangled representations, the number of latents available in the network structure does need to be at least as large as the number of true factors of variability in the generative process: if fewer are available, the network will necessarily either fail to learn some factors, or will entangle information about multiple factors into a single latent. In our simulations we have typically chosen a number of latent variables that was about double the number that we expected the fit model to need. While we have not explored this formally, informal experiments allowing larger numbers of latent variables resulted in networks that ultimately converged to similar solutions, though at the expense of longer training time and larger dataset size requirements.

---

> > ### Comment · Reviewer_Zh1M · 2023-08-15
> >
> > I thank the Authors for detailed response. I appreciate the addition of decision-making via accumulation of noisy evidence task. I updated my score accordingly.

---

### Official Review · Reviewer_SAvA · 2023-07-14

**Soundness:** 3 good
**Presentation:** 2 fair
**Contribution:** 3 good
**Rating:** 7
**Confidence:** 4

**Summary:**

The authors introduce a novel recurrent architecture that potentially learns more interpretable strategies than an LSTM while achieving roughly similar performance in fitting. They investigate the performance of their model on three separate datasets (two synthetic and one rat behavioral dataset), and offer qualitative evidence that the strategies learned by the model are easy to read out by first finding the "open bottlenecks," then looking at their activities. This is an exciting direction and I found the work very interesting and appropriate for NeurIPS.

**Update**

Increasing my score due to the updates provided by the authors. This is a very intriguing paper and a refreshing approach that could be especially useful for Neuro/Cog scientists.

**Strengths:**

- I like the motivation. The algorithmic approach is also very interesting as an extension of Beta-VAEs to RNNs.

- This is really cool how the open vs. closed-ness of bottlenecks is indicative of whether or not units are being used for solving a task! Sorry for the naive question, but would this provide more insight than if you were to simply regularize units for sparsity? Can you clarify on what this would tell above and beyond that?

**Weaknesses:**

- I think this is a potentially fantastic direction for cog and neuroscience and interpretability. I just wish there was more validation of the proposed architecture, and stronger evidence that the learned strategies are representative of those used by the rats in that dataset + that the model achieves similar performance as LSTMs as task complexity scales.

- The figures need work. Graphics are too small, there's awkward layouts, and color schemes need more contrast. I will list some detailed comments below.

- I find the Architecture figure very difficult to understand. I have faced the daunting task of making RNN figs in the past so believe me I understand the challenge here. But I don't find the relationship between left and right panels intuitive, understand how this could be used for computing, or what the goal is of this architecture. Just a thought, but maybe something more high-level? Or even consider removing this fig as you do a good job of explaining the model in the text.

- Fig 2b is hard to see. Could you stack the two parts of A on top of each other, then expand B?

- Figure 3 is difficult to understand. The text describing this is super cool and makes sense. But I don't get much from this figure. It feels more like SI to me.

- Figure 4, everything is too small. Am I correct in that 4C/G (and 3C/G for that matter) are cartoons of the data generating process? There's no clear correspondence between them and the data. I would either remove them or figure out a way to make the correspondence clearer with the data.

- Figure 5, The light orange and light blue are hard to read. Also this could be bigger.

- Figure 6, This one describes the most interesting dataset in the paper, but is hardest of all to read! I also don't understand whats going on by looking at it. I see the bottlenecks are changing as you reduce beta. But what does that mean? I know this is a tough ask but you could maybe focus on just one of these models and make it very intuitive to show how it is revealing an insight into the animals' strategies for solving the task.

- I am glad the authors brought up Ji et al in their related work. But I am also confused why they didn't compare to that approach. I assume this work is in progress? If these are networks with a "very small number of hidden units" as the authors wrote then it should be straightforward to do.

- The biggest limitation I see here is that the interpretability work is all postdictive. The model is used to explain existing generated datasets which is super cool, then fit to an animal dataset, which is even cooler. But the interpretability of this fit is purely qualitative. This would be a slam dunk if it could be validated experimentally in animal or human neural recordings (predicting neural activity after fitting to behavior) or behavioral data (e.g., identifying and testing biases, potentially).

**Questions:**

- Can you reduce the number of hidden units in the LSTM to roughly match the number of open bottlenecks in the DisRNNs, and get similar performance + interpretability? For interpretability, I guess you could look at the cell state of the circuit? It may not work of course because of the complexity of LSTMs.

- Is there a code release?

**Limitations:**

See weaknesses. This is tantalizingly close to a great paper. More evidence is needed.

---

> ### Author Rebuttal · Authors · 2023-08-10
>
> **Weaknesses**
>
> > The figures need work. Graphics are too small, there's awkward layouts, and color schemes need more contrast. I will list some detailed comments below.
>
> Thank you for these detailed and helpful suggestions!
>
> > I find the Architecture figure very difficult to understand. I have faced the daunting task of making RNN figs in the past so believe me I understand the challenge here. But I don't find the relationship between left and right panels intuitive, understand how this could be used for computing, or what the goal is of this architecture. Just a thought, but maybe something more high-level? Or even consider removing this fig as you do a good job of explaining the model in the text.
>
> We have substantially revised this figure, and believe that it is now much more clear (rebuttal pdf, top row). We welcome additional thoughts and suggestions.
>
> > Fig 2b is hard to see. Could you stack the two parts of A on top of each other, then expand B?
>
> We have increased the size of this figure
>
> > Figure 3 is difficult to understand. The text describing this is super cool and makes sense. But I don't get much from this figure. It feels more like SI to me. Figure 4, everything is too small. Am I correct in that 4C/G (and 3C/G for that matter) are cartoons of the data generating process? There's no clear correspondence between them and the data. I would either remove them or figure out a way to make the correspondence clearer with the data.
>
> We have increased the size of these figures. We have removed these cartoons. We agree that they don’t add much that is not already present in the “bottlenecks” plot.
>
> > Figure 5, The light orange and light blue are hard to read. Also this could be bigger..
>
> We have increased the contrast and the size of this figure
>
> > Figure 6, This one describes the most interesting dataset in the paper, but is hardest of all to read! I also don't understand whats going on by looking at it. I see the bottlenecks are changing as you reduce beta. But what does that mean? I know this is a tough ask but you could maybe focus on just one of these models and make it very intuitive to show how it is revealing an insight into the animals' strategies for solving the task.
>
> We have revised this figure and our description of it in the text, in an attempt to be more clear.
>
> > I am glad the authors brought up Ji et al in their related work. But I am also confused why they didn't compare to that approach. I assume this work is in progress? If these are networks with a "very small number of hidden units" as the authors wrote then it should be straightforward to do.
>
> We agree with the reviewer that this will be an important direction for future work. In informal exploration we have found that LSTMs with a very small number of hidden units typically underperform those with larger numbers of units. It will be important to compare directly using matched models (GRUs and their "S-GRU"), and datasets.
>
> > The biggest limitation I see here is that the interpretability work is all postdictive. The model is used to explain existing generated datasets which is super cool, then fit to an animal dataset, which is even cooler. But the interpretability of this fit is purely qualitative. This would be a slam dunk if it could be validated experimentally in animal or human neural recordings (predicting neural activity after fitting to behavior) or behavioral data (e.g., identifying and testing biases, potentially).
>
> We agree with the reviewer that a key future direction for this line of work will be to analyze in detail the fits of the disRNN to the laboratory datasets, and to relate them in detail to existing cognitive models as well as to neuroscientific data.
>
> **Questions**
>
> >Can you reduce the number of hidden units in the LSTM to roughly match the number of open bottlenecks in the DisRNNs, and get similar performance + interpretability? For interpretability, I guess you could look at the cell state of the circuit? It may not work of course because of the complexity of LSTMs.
>
> We informally explored this in previous work, and found that shrinking the LSTM down to just a few units typically results in a loss of predictive performance (the "best LSTM hyperparameters" used in figure 5 have 7-9 hidden units). We also informally explored fitting larger LSTMs and analyzing the dynamics of the first few principal components – this typically reveals retention of some dynamics that are not induced by training but instead leftover from network initialization. Frustration with these approaches is in fact what led us to the disRNN ideas of 1) explicitly penalizing the network for retaining information in its activations that it was not actually using for anything and 2) allowing the update rule for those activations to be free-form, parameterized by a feedforward sub-network, rather than constrained by particular update equations as in an LSTM or GRU.
>
> We will add new supplemental information investigating this in our datasets
>
> >Is there a code release?
>
> We promise to open-source our code and synthetic datasets as soon as possible, and definitely well in advance of the conference. The laboratory datasets we used are already publicly available from the authors of the papers in which they were originally reported.

---

> > ### Comment · Reviewer_SAvA · 2023-08-10
> > **Response**
> >
> > Thank you for your hard work in responding to my and the other reviewers' questions and critiques.
> >
> > I am disappointed however that you weren't able to compare to existing baselines as I suggested beyond offering the "informal" summary you provided here. I'm also disappointed that code is not available for us to look at to better understand your work. I believe that these are very basic model comparisons and requests for code (even just model architectures) that should not be difficult to complete in a day or two. Please advise if I am misguided in this.
> >
> > Could you fix these issues over this discussion period?

---

> > > ### Author Response · Authors · 2023-08-18
> > >
> > > Thanks for pushing us on this! We agree that the manuscript would be stronger if it also explored whether conventional neural network architectures might also be useful for cognitive model discovery. We have added new analysis using GRUs, rather than LSTMs, because related work has reported good performance on similar datasets (Ji et al., Dezfouli et al., etc), and because having just one type of recurrent unit makes them easier to analyze. We will add this analysis to our manuscript as supplemental information.
> > >
> > > 1. For each of the five datasets (synthetic Q-Learning, synthetic actor-critic, synthetic bounded accumulator, rats two-armed bandit, rats, poisson clicks), we have fit GRUs of different sizes and report cross-validated quality-of-fit. We find for the synthetic datasets that “tiny” GRUs with only one or two recurrent units do provide a quality of fit that is broadly similar to that of larger networks. For the rat datasets, we find that the best quality of fit comes from larger networks. These results are reported in the table below. In the submitted manuscript, we showed that disRNN can provide a similar quality of fit on the rat two-armed bandit datasets to that of larger RNNs (the “best LSTM” hyperparameters called for either eight or nine units).
> > > 2. For the three synthetic datasets, we have fit “tiny” two-unit GRUs, and examined plots of example sessions and update rules. We see that these typically do not have a 1:1 relationship with the true generative latent variables. The exception is the Q-Learning agent, for which two-unit GRUs do discover a disentangled solution. Solutions found for the other agents are fully entangled, with each unit’s update dependent on the value of the other unit and on both input variables. We interpret this to mean that very small conventional networks can learn dynamics that recapture those of certain generative processes, but that they do not do so reliably. (It does not look like I have the option to update my “rebuttal pdf” at this time to show you these figures -- please let me know if I’m wrong about this or if there is another way to share them!).
> > > 3. For each of the three synthetic datasets, we fit larger ten-unit GRUs and summarize their dynamics using the first two principal components. While some human-interpretable structure is visible, there is still not a 1:1 relationship between PCs and the latent variables of the generative process. The dynamics are entangled, with each PC’s update depending on all inputs and on the value of the other PC.  We interpret this to mean that the dynamics of the first few PCs of conventional networks can reveal human-interpretable structure, but do not reliably recapture the dynamics of the generative process.
> > >
> > > Taken together, we interpret these results to indicate that conventional neural networks like GRUs definitely can be a viable route to cognitive model discovery in some circumstances, but also that they have important limitations. One limitation is that, while task training ensures that the dynamics they contain are sufficient to solve the task, nothing ensures that all aspects of these dynamics are necessary (they are free to retain epiphenomenal dynamics). Another limitation is that, while the number of latent variables can be constrained by limiting network size or by only considering the top few PCs, nothing ensures that the update rules for these variables are sparse, and nothing encourages them to be “axis aligned”, mapping 1:1 onto the true generative dynamics.
> > >
> > > **Difference in Cross-Validated Normalized Likelihood vs Reference (Percentage Points)**
> > >
> > > Synthetic datasets: Average of three random seeds.
> > >
> > > Rat Two-armed Bandit dataset: Average of three random seeds for each of twenty rats
> > >
> > > Rat Poisson Clicks dataset: Average of three random seeds for each of nineteen rats
> > >
> > > |  | GRU1 | GRU2 | GRU3 | GRU4 | GRU5 | GRU6 | GRU7 | GRU8 | GRU9 | GRU10 | GRU11 | GRU12 | GRU13 | GRU14 |
> > > | ----------- | ----------- | ----------- | ----------- | ----------- | ----------- | ----------- | ----------- | ----------- | ----------- | ----------- | ----------- | ----------- | ----------- | ----------- |
> > > | Q-Learning | -0.72 | ref | 0.002 | 0.002 | 0.003 | 0.002 | 0.002 | 0.002 | 0.002 | 0.002 | 0.002 | 0.002 | 0.001 | 0.001 |
> > > | Actor-Critic | -1.38 | ref | 0.01 | 0.02 | 0.02 | 0.02 | 0.02 | 0.02 | 0.02 | 0.02 | 0.02 | 0.02 | 0.02 | 0.02 |
> > > | Bounded Accumulation | ref |-0.008 | 0.001 | 0.003 | 0.004 | 0.004 | 0.004 | 0.004 | 0.004 | 0.004 | 0.004 | 0.004 | 0.004 | 0.004 |
> > > | Rat Two-Armed Bandit | -2.16 | ref | 0.34 | 0.26 |  0.39 |  0.38 | 0.35 | 0.46 | 0.44 | 0.38 | 0.35 | 0.51 | 0.53 | 0.50 |
> > > | Rat PClicks | -0.36 | ref | 0.08 | 0.07 |  0.08 |  0.09 |  0.08 | 0.10 |  0.11 |  0.05 |  0.10 |  0.10 | 0.10 | 0.13 |

---

> > > > ### Comment · Reviewer_SAvA · 2023-08-18
> > > > **Response**
> > > >
> > > > Excellent, thank you for this! It is super helpful for understanding why your method is important.
> > > >
> > > > Could you also provide code?

---

> > > > > ### Author Response · Authors · 2023-08-19
> > > > >
> > > > > > Could you also provide code?
> > > > >
> > > > > I've looked into this, and I'm disappointed to say that it doesn't seem like there's anything I can do to accelerate the open-sourcing process. It will definitely be done in time for the conference, but will not be possible during the discussion period.
> > > > >
> > > > > It's important to us that the manuscript is clear enough that readers will be able to understand what we did without needing to refer to the open-source code (which is for those who might wish to reproduce or extend our work, and shouldn’t be necessary for most readers). If you have specific questions that would be answered by the code but aren't addressed in the manuscript, we'd love to know what they are, so we can improve the manuscript!

---

> > > > > > ### Comment · Reviewer_SAvA · 2023-08-19
> > > > > > **Response**
> > > > > >
> > > > > > Thank you for the response. I really appreciate all of the work during this response period.
> > > > > >
> > > > > > If the code will be made available I am satisfied. I'm increasing my score.

---

### Official Review · Reviewer_9K9A · 2023-07-23

**Soundness:** 2 fair
**Presentation:** 2 fair
**Contribution:** 2 fair
**Rating:** 5
**Confidence:** 4

**Summary:**

The authors propose a methodology to learn (or system-identify) parsimonious cognitive models directly from data. Specifically, they introduce the idea of disentangled RNNs (DisRNNs). DisRNNs are gated recurrent neural networks with additional bottleneck constraints. These (learnable) bottlenecks bound the information transfer capacity by parametrically controlling the signal-to-noise ratio per latent unit. Moreover, they propose that each latent unit in the DisRNN is updated independently per its own learning rule. The authors demonstrate the effectiveness of DisRNNs on a dynamic two-arm bandit task using synthetic and animal behavioral data.

**Strengths:**

The authors raise important points about data-driven models usually being inscrutable. System identification is a long-standing problem of interest over the years that several studies have aimed to address. Combining system identification with data-driven approaches is a certainly promising direction. This information-theoretic approach to controlling signal quality through noise induction is interesting. Finally, though speculative, the authors provide a sense of how testable hypotheses for neuroscience can be obtained from their model.

**Weaknesses:**

My main concern about this manuscript is its limited scope (in the formulation and experiments). The authors motivate their approach by stating that "discovering an appropriate model structure ..." (L21 Pg. 1). However, the model structure of the DisRNN has several carefully chosen inductive biases that closely mimic the model tested in this paper (including the linear update terms and logistic readout). The fact that the DisRNN is able to learn exact parameter specifications is not necessarily surprising. The interpretability aspect of the learned DisRNN also relies heavily on the apriori known ground truth model. Can the authors test this model on other cognitive process models, even within the scope of decision-making?

The way the DisRNNs are set up, it is unclear to me how they'll scale -- both as a function of the complexity of the underlying model governing data and in terms of parameter and sample complexity. Particularly since each latent dimension has its own associated MLP parameters, it will be very beneficial for the manuscript to have extended numerical evaluations on this front.

Writing style:
The general clarity of the article in a few places can be improved. Here are some of my suggestions. I'd encourage the authors to pay more attention to language of this kind throughout the manuscript.

Fig. 1 must be improved. The arrow marks (and corresponding colors) do not have a clear legend. The orange text indicates that these lines are modulated by a bottleneck but what does the intersection of orange and blue lines mean? Similarly, the "Update rule X" can be depicted as an MLP while clearly showing the dimensionality of $z$. It is unclear if the indices authors are using denote time (which seems to be the case for observations) but not for $z$.

Clarity: "In order to learn an interpretable cognitive model, we encourage sparsity." Can the authors expand and clarify why sparsity implies interpretability?

Clarity: "Synthetic datasets from two reinforcement learning agents performing this task.." It's perhaps better to say that data was generated using ground truth update equations since the "agents" themselves here were not trained but rather specified.

**Questions:**

Please refer to the weaknesses section above.

**Limitations:**

Please refer to the weaknesses section above.

---

> ### Author Rebuttal · Authors · 2023-08-10
>
> **Weaknesses**
> > My main concern about this manuscript is its limited scope (in the formulation and experiments). The authors motivate their approach by stating that "discovering an appropriate model structure ..." (L21 Pg. 1). However, the model structure of the DisRNN has several carefully chosen inductive biases that closely mimic the model tested in this paper (including the linear update terms and logistic readout). The fact that the DisRNN is able to learn exact parameter specifications is not necessarily surprising. The interpretability aspect of the learned DisRNN also relies heavily on the apriori known ground truth model. Can the authors test this model on other cognitive process models, even within the scope of decision-making?
>
> We thank the reviewer for pointing out this concern. We agree that the multiplicative update rules used in the Q-Learning agent and by the Actor component of the Actor-Critic are very similar to the multiplicative update we built into disRNN, and that their logistic decision rule is similar in form to the softmax cross-entropy loss function. We have now added a new set of experiments training the same disRNN architecture on synthetic datasets generated by a bounded accumulation agent. This agent shares neither of these features: the update rule for its decision variable is additive, with a nonlinearity at the sticky bound; its choice rule is binary with a lapse rate. We find that disRNN is able to recover the structure of this agent.
>
> We hope that this goes some way to reassure the reviewer that the structure of disRNN is suitable for fitting generic dynamical systems. It is of course also possible to construct disRNN using additive update rules $ z_i^{t+1} = z_i^t + \\text{MLP}_i (\\mathbf{z}^t, \\mathbf{o}^t) $, which more-obviously express generic dynamical systems. In informal exploration we found that these required more time to train but ultimately converged on similar solutions. Exploring more rigorously whether there are advantages to one variant or the other may be a useful question for future work.
>
> > The way the DisRNNs are set up, it is unclear to me how they'll scale -- both as a function of the complexity of the underlying model governing data and in terms of parameter and sample complexity. Particularly since each latent dimension has its own associated MLP parameters, it will be very beneficial for the manuscript to have extended numerical evaluations on this front.
>
> We agree with the reviewer that discovering cognitive models that are more complex, for example using behavior from more complex tasks, will likely require larger networks and therefore larger datasets, and that it seems likely that this will prove impractical for some applications, at least without introducing additional regularization. We note that in cognitive neuroscience, a large fraction of the literature focuses on tasks similar in complexity to those we examine here. These tasks are chosen because they are thought to isolate, in an experimentally tractable way, key building blocks of cognition. We believe that disRNN and methods like it will be useful for more complex tasks. But even if they are not, accelerating discovery in domains like trial-by-trial reward learning and like decision-making is in itself an impactful contribution. We have expanded the discussion of these issues in our manuscript.
>
> > Fig. 1 must be improved. The arrow marks (and corresponding colors) do not have a clear legend. The orange text indicates that these lines are modulated by a bottleneck but what does the intersection of orange and blue lines mean? Similarly, the "Update rule X" can be depicted as an MLP while clearly showing the dimensionality of z. It is unclear if the indices authors are using denote time (which seems to be the case for observations) but not for z.
>
> We have improved this figure following these suggestions, and those of the other reviewers (rebuttal pdf, top row).
>
> > Clarity: "In order to learn an interpretable cognitive model, we encourage sparsity." Can the authors expand and clarify why sparsity implies interpretability?
>
> We thank the reviewer for calling our attention to this important point. We have expanded our discussion to address it in detail. The relevant passage now reads:
>
> *Limiting the number of latent variables provides three distinct benefits. The first is that such a model is more likely to be useful for scientific tasks, such as searching for correlates in measurements of neural activity, that involve interacting with finite datasets. The second is that interpreting a fit disRNN requires a human expert to inspect the update rules. The smaller the number of latents and fewer inputs to the update rule for each, the less cognitive burden will be placed on that human expert, and the more likely they will be able to arrive at a satisfying human intuition about the cognitive mechanism embodied by the model. The third is that goal of discovery is to identify models that human experts will consider to be cognitively plausible. When evaluating classic handcrafted models, many experts agree that, all else being equal, simpler models (smaller number of equations, fewer terms in each equation) are more plausible.*
>
> > Clarity: "Synthetic datasets from two reinforcement learning agents performing this task.." It's perhaps better to say that data was generated using ground truth update equations since the "agents" themselves here were not trained but rather specified.
>
> We thank the reviewer for pointing out this issue of vocabulary. In cognitive science, the term “agent” is frequently used to denote software modules that take “actions”, interacting in closed-loop with an “environment”, regardless of whether these are hand-crafted or themselves the result of machine learning. We do feel this usage is appropriate here, but have added clarifying language to our manuscript in several places to assist readers from other backgrounds.

---

> > ### Comment · Reviewer_9K9A · 2023-08-21
> > **Appreciate the extensive response**
> >
> > I thank the authors for their efforts in responding to mine as well as other reviewer's comments in detail.
> >
> > > Bounded accumulator
> >
> > Thanks for these experiments. Is there a reference for this process model? It would be good to see the exact ground truth update equations here. In general, I do agree that it's seemingly different from the Q-learning and the A-C agents. The fact that the disRNN is able to fit data interpretably from this model is a good sign.
> >
> > > About scaling
> >
> > Thanks for the comment. If I may clarify, my point about scaling was not only about extending this to large-scale problems. More so that the focus on "interpretability" here heavily relies on knowing the models under consideration apriori. This issue was raised in my initial review as well. I agree with the authors that most of the cognitive neuroscience literature has focused on models of this size. However, it is unclear to me and I would appreciate it if the authors can clarify how the methodology proposed here can **accelerate discovery** on the interpretable modeling front, for unknown systems.
> >
> > > The terminology "agent"
> >
> > If that's a term of art, I can understand the usage of the word "agent". In which case, as a reader, I would find it much clearer if the "reinforcement learning" prefix was dropped since this implies some form of learning/training.
> >
> > Overall, I do appreciate the authors' efforts during the discussion period and though I still reserve some concerns, I am happy to update my evaluation.

---

### Official Review · Reviewer_M5Mx · 2023-07-26

**Soundness:** 4 excellent
**Presentation:** 4 excellent
**Contribution:** 3 good
**Rating:** 7
**Confidence:** 3

**Summary:**

This study introduces "Disentangled RNNs" (DisRNNs), a type of interpretable RNN designed with sparse latent variables and simple update rules. The model's interpretability is achieved by utilizing noisy channels both to maintain the state of each latent variable and to read out these states into the update rules. An auxiliary loss function penalizes channels with non-zero Signal-to-Noise Ratio, balancing model fit with sparseness and simplicity.

The authors trained the DisRNNs directly on behavioral data and then examined the resulting latent representation and learned update rules. First, the authors trained DisRNNs on synthetic behavioral data sampled from either a Q-learning agent or an actor-critic agent, successfully replicating the generating latent variables and their corresponding update rules. Subsequently, the authors trained DisRNNs on real behavioral data from mice performing a dynamic two-armed bandit task. The DisRNNs achieved a competitive cross-validated fit to the behavioral data. Upon analysis, a DisRNN fitted to a specific mouse exhibited a strong resemblance to the best-known human-derived model, with the model's complexity varying based on the weight assigned to the auxiliary loss.

**Strengths:**

1. This manuscript presents a compelling approach to interpretable machine learning. The use of noisy channels is well justified from a computational neuroscience perspective, and the validation of the model on synthetic data from artificial agents is logically sound. Overall, this work may provide a promising foundation for further research in neuroAI using interpretable models.

2. The manuscript is exceptionally well-written. The foundational principles of the reinforcement learning agents are elucidated with the clarity of an outstanding textbook, and the algorithm's description is concise yet clear and accessible.

**Weaknesses:**

1. Upon reading the paper, I found it somewhat disappointing that the interpretation of the DisRNNs trained on real behavioral data was quite cursory, focusing on a "typical-best" example without a systematic analysis of the learned representations across the trained DisRNNs. The study would have been more comprehensive and impactful had it demonstrated how applying this method to behavioral data could facilitate neuroscientific discovery.

2. The manuscript lacks explicit details regarding the selection of the parameter $\beta$ in section 4 (i.e., experiments with synthetic data). Consequently, the robustness of the procedure to the choice of $\beta$ remains unclear; it is uncertain whether there was a need for considerable adjustment of $\beta$ to achieve a good fit or to yield a sensible latent representation. This omission, coupled with the absence of open-source code and data, somewhat undermines my confidence in the applicability of this approach "out of the box".

**Questions:**

What should guide the selection of $\beta$ for an unknown system? Is it appropriate to determine $\beta$ based on cross-validated fit? In essence, how can we interpret the continuum of models obtainable along the bias-complexity tradeoff?

**Limitations:**

The limitations are adequately discussed.

---

> ### Author Rebuttal · Authors · 2023-08-10
>
> **Weaknesses**
> > Upon reading the paper, I found it somewhat disappointing that the interpretation of the DisRNNs trained on real behavioral data was quite cursory, focusing on a "typical-best" example without a systematic analysis of the learned representations across the trained DisRNNs. The study would have been more comprehensive and impactful had it demonstrated how applying this method to behavioral data could facilitate neuroscientific discovery.
>
> We agree with the reviewer that a key future direction for this line of work will be to analyze in detail the fits of the disRNN to the laboratory datasets, and to relate them in detail to existing cognitive models as well as to neuroscientific data. In the current manuscript, we hope to establish that applying this method to behavioral data results in models that are suitable for standard neuroscientific workflows that currently rely on handcrafted models. For example, the timecourses of their latent variables might be used as regressors for analyzing neural activity, or they might be run to generate synthetic datasets that make predictions about animal behavior in new situations. We will revised the manuscript to be more clear about this.
>
> > The manuscript lacks explicit details regarding the selection of the parameter Beta in section 4 (i.e., experiments with synthetic data). Consequently, the robustness of the procedure to the choice of Beta remains unclear; it is uncertain whether there was a need for considerable adjustment of Beta to achieve a good fit or to yield a sensible latent representation. This omission, coupled with the absence of open-source code and data, somewhat undermines my confidence in the applicability of this approach "out of the box".
>
> We have added an analysis sweeping a range of $\\beta$s for each of our synthetic data fits (rebuttal pdf, bottom row). We find that good predictive fit can be found over a wide range of values of $\\beta$, and that sensible latent representations can be found at the largest values of $\\beta$ which also produce good fit. We will include this analysis in our revised manuscript, and also revise the text to be more clear about this.
>
> We promise to open-source our code and synthetic datasets as soon as possible, and definitely well in advance of the conference. The laboratory datasets we used are already publicly available.
>
> **Questions**
> > What should guide the selection of Beta for an unknown system? Is it appropriate to determine based on cross-validated fit? In essence, how can we interpret the continuum of models obtainable along the bias-complexity tradeoff?
>
> For our synthetic datasets, ground-truth involved only a small number of latent variables, and a procedure of selecting from among the models with good predictive performance the one with the smallest number of open bottlenecks would have been sufficient to identify it. For discovering cognitive models using laboratory datasets, we expect that the best $\\beta$ will depend on the scientific use-case of the model, and that it may sometimes be useful to consider a model which does not achieve the best cross-validated fit, for example because the disRNN has been fit to a large behavioral dataset, but its latents are being used as regressors for a much dataset of neural recordings.
>
> Although this does mean that disRNN can produce multiple models from the same dataset, it is worth noting two things: 1) Over several orders of magnitude of beta, it is typical that only a handful of distinct models emerge, and they are often closely related, for example with one new latent appearing as beta is reduced. 2) These models can often be thought of as different levels of resolution on the cognitive mechanism. Each can be useful depending on the level of resolution needed for the research question.
>
> If the fit disRNN is to be used as a cognitive model, a key question is whether the mechanistic claims it embodies are psychologically and biologically plausible about the system being studied. Evaluating this necessarily requires considering not only predictive performance and model simplicity, but also expert domain knowledge. We expect that disRNN will be most useful in a workflow that involves fitting different networks to discover a number of different models,  then applying domain expertise as a filter to identify which (if any) are cognitively plausible and practically useful for the user’s current scientific goals.
>
> We have expanded our discussion of these issues in the manuscript.

---

> > ### Comment · Reviewer_M5Mx · 2023-08-15
> >
> > I thank the authors for their detailed reply. I have no further questions. However, I find that [the remaining questions from "SAvA"](https://openreview.net/forum?id=SOEF0i0G1z&noteId=IILRgByF3Z) are significant, and I look forward to your replies.

---

> > > ### Author Response · Authors · 2023-08-19
> > >
> > > We have responded to the questions from reviewer SAvA below. If you have additional follow-up questions please do let us know.

---

### Author Rebuttal · Authors · 2023-08-09

We thank the five reviewers for their insightful and helpful feedback on our manuscript. Three major concerns stood out to us as raised in similar ways by multiple reviewers. We summarize our responses to these three concerns here. We have also responded separately to each reviewer’s individual concerns.

1) Reviewers felt that the manuscript did not sufficiently make the case that disRNN is likely to be useful across a variety of different cognitive neuroscience tasks. They raised the possibility that it might contain structural biases that tailor it specifically to the domain of dynamic reward-learning tasks.

    We have addressed this concern by adding new experiments testing disRNN in a very different domain: decision-making via accumulation of noisy evidence. This domain, like that of reward-learning, is heavily studied in behavioral neuroscience and has been the subject of intensive cognitive modeling efforts. We consider specifically the “Poisson clicks” task (Brunton, Botvinick, and Brody, 2013). In each trial of this task, rats are presented with a series of auditory clicks delivered from a pair of speakers, one to the left and one to the right of the rat, and they are rewarded for reporting which speaker delivered a larger number of clicks.

    We first considered a synthetic behavioral dataset (click times and choices) generated by a “bounded accumulator” agent performing this task (rebuttal pdf, second row). This agent keeps track of the relative number of clicks on each side, and commits to a decision when the absolute difference crosses a bound, ignoring any clicks that occur after this bound has been crossed. We fit a copy of DisRNN to this synthetic behavioral dataset, and find that it is able to recover the structure of the bounded accumulator agent.

    We then considered an open-source laboratory dataset from rats performing this task (Brunton, Botvinick, and Brody, 2013). We fit several copies of DisRNN to behavioral data from individual rats, using a variety of different complexity penalties. We find that these often recover human-interpretable models which capture known features of rat behavior on the task. In the examples shown (rebuttal pdf, third row), the green latent is tracking the relative number of clicks on each side, while the blue latent is tracking a short-term sensory adaptation effect (clicks that shortly follow another click have a reduced impact on decision-making) that is known to play an important role in this task.

2) Reviewers wondered how robust the fits to synthetic data were with respect to various hyperparameters, especially the information penalty \beta.

    We have added simulations sweeping this hyperparameter in each of the three synthetic datasets (Q-Learning, Leaky Actor-Critic, and Bounded Accumulation) and report both cross-validated quality-of-fit and the number of open information bottlenecks (rebuttal pdf, bottom row). We find in each case that the true model structure (dotted lines) is reliably recapitulated by the networks which are the simplest (smallest number of open bottlenecks) that also achieve good predictive performance (comparable to that of the best models).


3) While reviewers felt that the clarity of the writing was high, they felt that the clarity of the figures could be improved, especially Figure 1, which is critical for readers to understand as it explains the structure of the model.

    We have substantially improved this figure (rebuttal pdf, top row) and made a number of edits to the other figures as well, relying heavily on the detailed and generous advice of reviewer SAvA. We believe this has greatly improved the clarity of the figures and of the manuscript as a whole.

---

### Decision · Program_Chairs · 2023-09-21

**Decision:**

Accept (poster)

**Comment:**

There was general agreement that the work described is significant for the brain sciences and for X-ai. The reviewers raised some concerns regarding the experimental validation of the approach including more substantial evidence that the learned strategies are representative of those used by animals. Overall, many of the initial concerns seem to have been well addressed in the rebuttal. The paper received moderate but unanimous support from the reviewers. The AC recommends the paper to be accepted.